behaviour, ecology, evolution

migration, satellite, tracking, flight performance, orientation, weather

**Author for correspondence:**
Bart Kempenaers
e-mail: b.kempenaers@orn.mpg.de

# Wind conditions influence breeding season movements in a nomadic polygynous shorebird

Johannes Krietsch, Mihai Valcu and Bart Kempenaers

Department of Behavioural Ecology and Evolutionary Genetics, Max Planck Institute for Ornithology, Eberhard-Gwinner-Strasse, 82319 Seewiesen, Germany

JK, 0000-0002-8080-1734; MV, 0000-0002-6907-7802; BK, 0000-0002-7505-5458

Nomadism is a behaviour where individuals respond to environmental variability with movements that seem unpredictable in timing and direction. In contrast to migration, the mechanisms underlying nomadic movements remain largely unknown. Here, we focus on a form of apparent nomadism in a polygynous shorebird, the pectoral sandpiper (*Calidris melanotos*). Local mating opportunities are unpredictable and most males sampled multiple sites across a considerable part of their breeding range. We test the hypothesis that individuals decided which part of the breeding range to sample in a given season based on the prevailing wind conditions. Using movement data from 80 males in combination with wind data from a global reanalysis model, we show that male pectoral sandpipers flew with wind support more often than expected by chance. Stronger wind support led to increased ground speed and was associated with a longer flight range. Long detours (loop-like flights) can be explained by individuals flying initially with the wind. Individuals did not fly westwards into the Russian Arctic without wind support, but occasionally flew eastwards into the North American Arctic against strong headwinds. Wind support might be less important for individuals flying eastwards, because their autumn migration journey will be shorter. Our study suggests that individuals of a species with low site fidelity choose their breeding site opportunistically based on the prevailing wind conditions.

## 1. Introduction

Flying or swimming animals move within a medium which is in motion itself. This means that the animal's trajectory is the result of their own speed and heading, and the speed and direction of the flow of the medium [1]. In birds, a growing body of literature describes the substantial effects of wind on flight, mainly within the framework of 'optimal migration' [2–4]. Migratory birds can optimize their flight efficiency (i.e. time in flight, flight range and energy expenditure) in relation to wind by (i) selecting a departure time with the most favourable winds [3,5,6], (ii) choosing a flight altitude with the most favourable winds [7–9], (iii) choosing an optimal flight speed and mode (i.e. flapping, gliding or soaring) [10–13], and (iv) adjusting their route to use the most favourable winds [14–17]. Whether and how individuals can use the wind during flights depends on species-specific differences in life history and ecology that allow different degrees of variability regarding timing of movement and space use [18–23]. Most migratory birds show high philopatry, returning to the same breeding, stopover and wintering sites each year [24]. Some species also have highly repeatable routes, but can be flexible in their timing of migration (e.g. [16]), while other species adjust their route based on wind conditions, despite being constrained in time [18–20]. For individuals flying towards a specific goal, the use of optimal wind conditions might be

constrained or overridden by other factors. For example, individuals might face trade-offs between waiting for favourable winds and optimal timing [2,21,22], or selection might favour individuals that avoid ecological barriers or cross them on the shortest route, independently of the wind conditions. Ultimately, these factors may define whether individuals benefit from being flexible or consistent with respect to timing and route (e.g. [18]), and consequently determine whether consistency or plasticity in movements are favoured by selection.

Here, we consider wind use during a recently discovered form of nomadic movement in a species that shows extremely low site fidelity, the pectoral sandpiper (*Calidris melanotos*). Pectoral sandpipers are polygynous and males compete for access to fertile females, but provide no parental care [25,26]. In spring, individuals migrate from their wintering grounds in the Southern Hemisphere (mostly South America) to the Arctic. However, most males do not stay at one breeding site. Presumably in response to variation in local mating opportunities, males sample several potential breeding sites during the four to six weeks lasting breeding season, often throughout a considerable part of the species's breeding range, covering distances of up to 13 000 km [27]. Individuals can move between successive sites in any direction, at any time and over variable distances [27,28]. The low local return rates of females (less than 1% [29]) and large between-year variation in snow accumulation and timing of snowmelt in the Arctic [30] result in locally unpredictable mating opportunities [27] and presumably in substantial spatio-temporal variation in breeding opportunities throughout the species's breeding range.

Given that local mating opportunities are unpredictable [27], that local competition is intense and presumably energetically costly [31] and that time is an important resource (short breeding season in the Arctic), males may use wind support to move to the next site quickly and with the least energy expenditure. Because the quality of a distant site is unpredictable, the costs associated with reaching a particular site should be important for individuals making decisions about where to go next. Wind patterns over the Arctic Ocean are highly variable between years and within a season [32]. Thus, the aim of our study is to investigate (i) whether individual pectoral sandpipers use the local wind conditions to decide in which part of the breeding range they will sample potential breeding sites and (ii) how the wind conditions en route influence their flights.

We used movement data from 80 males that departed from a breeding site in the centre of their breeding range. Males flew either west into the Russian Arctic, or east into the North American Arctic, but the proportion going west differed between years [27]. Using wind data from a global reanalysis model [33], we first assessed how wind conditions influenced the males' flight. We characterized the wind conditions connected to the flights and estimated the most likely flight altitude. We then analysed the effect of wind support on ground and air speed, predicting that increasing wind support resulted in both faster ground speeds (i.e. reduced flight time) and lower air speeds (i.e. reduced flight costs) [34,35]. Second, we analysed whether the direction of male movements was associated with wind conditions within and between years. We examined whether the local wind direction and speed influenced the males' initial flight direction. We then asked if the initial flight direction predicted where individuals settled next. We also tested whether individuals timed their departure to make use of optimal wind support, assuming that their final destination was their goal. Finally, we compared the wind support on the actual track with that on the shortest route to the destination to assess whether the use of wind support can explain the observed large detours (loop flights).

## 2. Material and methods

### (a) Tracking data
In both 2012 and 2014, 60 adult male pectoral sandpipers were caught and equipped with 5 g Solar Argos PTT-100 (Microwave Telemetry Inc.) satellite transmitters near Utqiaġvik (Barrow), Alaska (71°18′ N, 156°44′ W) between 25 May and 7 June. All transmitters had a continuous duty cycle and we obtained on average 2.8 positions per hour for the flights analysed here. Raw Argos data were filtered and a continuous time-correlated random walk model was used to predict maximum-likelihood locations every 15 min. For further details about tag attachment and data processing, see [27]; the complete dataset is available at http://dx.doi.org/10.17605/osf.io/vx2mk.

For this study, we selected a subset of tracks with the following criteria: (i) departure date before 8 July (excluding movements after the latest clutch initiation date, based on [36]), (ii) departure location less than 250 km from Utqiaġvik, (iii) departure track over the Chukchi or Beaufort Sea (excluding four over-land tracks), and (iv) track length greater than 500 km (figure 1). Because males first flew a considerable distance over sea (ice), we can exclude the possibility that they were assessing conditions on land (e.g. snow cover) to decide where to settle next. In total, 85 tracks (2012: $n = 49$; 2014: $n = 36$) from 80 individuals fulfilled these criteria and were used for further analyses. In both years, the area covered by the flight paths was almost entirely covered with sea ice (electronic supplementary material, figure S1). Thus, we can exclude that variation in ice cover played a role in decisions about where to go.

Five males are included twice, because they flew two tracks within a season that fitted our selection criteria. Four of them made a loop flight (see below), followed by a directional flight, while one first flew east, then west. A sixth male made a loop flight, followed by a directional flight that would have fitted the selection criteria, except that the departure location was greater than 250 km from Utqiaġvik.

### (b) Track description
For each track, we defined the following variables. (i) Departure time: time of the first position over the ocean. Males mainly left during the 'night'. Thus, for analyses that included departure day, we centred the days around midnight by subtracting 12 h from each day, such that all birds that left in one night were counted as belonging to one departure 'day'. (ii) Arrival time: time of the last position over the ocean. We excluded positions over land until the next residency area (identified as spatial clusters of points; for details see [27]) was reached, because the environmental cues might change in comparison to those while flying over the ocean. In most cases, the distance from the coast to the residency area was small relative to total track length (median: 73 km, range: 1–1147 km). (iii) Flight time: the period between departure and arrival; birds did not rest on the sea ice [27]. (iv) Track length: the sum of the distances between all consecutive positions between departure and arrival. (v) Track detour: the difference between track length and the shortest distance between departure and arrival location. (vi) Percentage detour: track detour divided by track length. (vii) Track straightness: track length divided by the shortest distance between the departure and arrival location [37]. A straightness of 1 indicates a direct flight between two points.

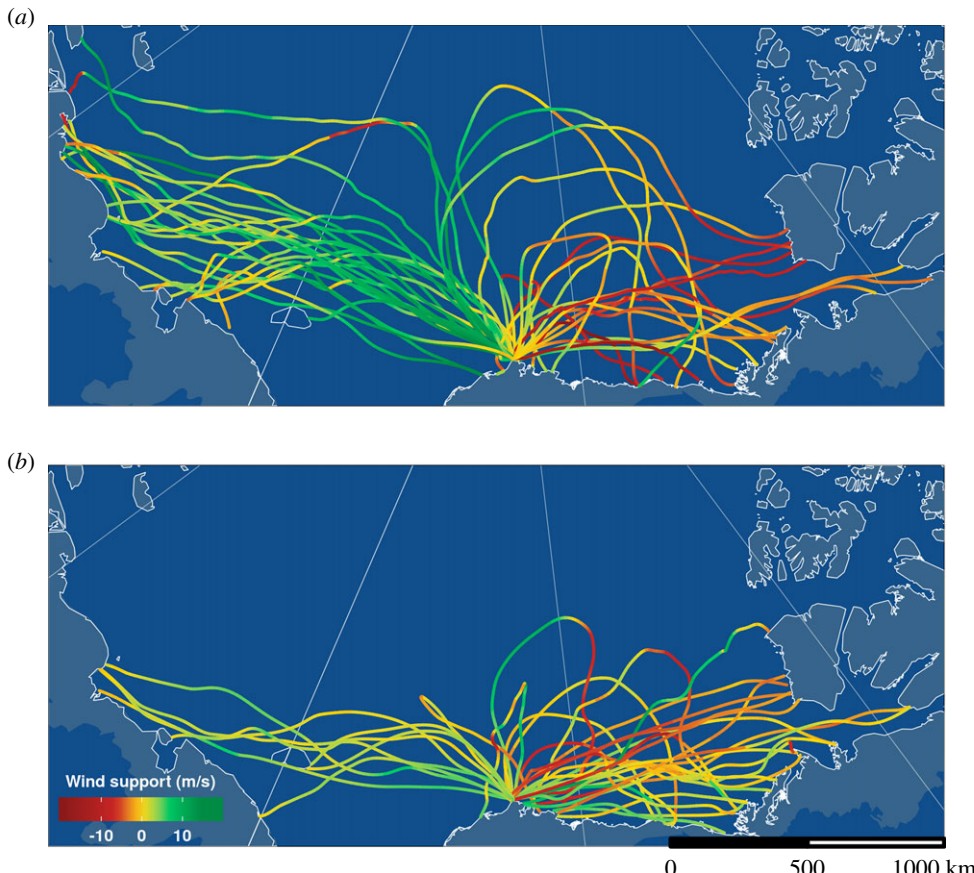

**Figure 1.** Tracks of male pectoral sandpipers that left Utqiaġvik, Alaska (lower centre of the map; capture site) during the breeding season between 30 May and 23 June in 2012 (*a*; $n = 49$) and 2014 (*b*; $n = 36$). Track colour indicates wind support, i.e. the length of the wind vector in the direction of the bird's movement (ground vector), calculated using wind data at approximately 750 m altitude (see Material and methods). Tailwinds are shown in green, headwinds in red. The grey-blue area indicates suitable breeding habitat within the known breeding range of the species [27]. Map projection: polar Lambert azimuthal equal area with longitude origin 156.65° W (Utqiaġvik) from Natural Earth (http://www.naturalearthdata.com). For a movie of these flights and for individual tracks see electronic supplementary material (movie available at https://www.youtube.com/watch?v=A-Q5J1wRBUA&feature=youtu.be). (Online version in colour.)

## (c) Wind data

We used wind data from the ECMWF ERA-Interim reanalysis [33] with a spatial resolution of approximately 80 km/0.75° and a temporal resolution of 6 h. We extracted the *u*- (west–east) and *v*- (south–north) wind components for six altitudes: 10 m above ground and at the pressure levels 1000, 925, 850, 775 and 700 hPa, roughly corresponding to 100, 750, 1500, 2250 and 3000 m.a.s.l. (metres above-sea level). The actual flight altitudes are unknown, but we chose these altitudes because radar data from the region indicate that shorebirds (including pectoral sandpipers) mainly used altitudes between 0 and 3000 m.a.s.l. [38]. Data were resampled with a bicubic interpolation to a spatial resolution of 10 km. To test whether our results are sensitive to the time scale of the wind data, we shifted all tracks ±6 h and calculated Pearson product–moment correlation coefficients comparing the mean wind support for each altitude (see below). Shifted and observed data correlated strongly (all $r > 0.98$, $n = 85$). Thus, the temporal resolution of the wind data is adequate. To assess the effect of wind on departures at a finer temporal scale, we used local wind data from the weather station in Utqiaġvik (https://www.esrl.noaa.gov). These data contain hourly averages of wind speed and direction at ground level.

## (d) Wind conditions in the area of the flights

We defined the area used by the males in our dataset during their flights over the ocean as a convex hull covering all tracks and excluding all land surface. Based on the known breeding range and suitable habitat of the species [27], we defined the

corresponding area over land the birds could have used to move to the same site (with the exception of crossing the Bering Strait). For both areas ('over ocean' and 'over land'), we calculated daily mean wind speeds and directions for the period of actual departures.

## (e) Wind support, relative wind support and crosswind

For each individual, we calculated ground speed and direction of movement (0° as north) between subsequent points on its track. Using wind data (at the six altitudes) closest in space and time to each position, we calculated wind support as the length of the wind vector in the direction of the bird's flight (ground vector) as described in [11]. Positive values represent tailwinds, negative values headwinds.

To quantify how birds used the wind independently of wind speed, we calculated 'relative wind support' as wind support divided by wind speed. Values range between −1, for birds that fly in the exact opposite direction to the wind (headwind), and +1, for birds that fly exactly in the direction of the wind (tailwind).

We calculated the crosswind component as the length of the wind vector perpendicular to the ground vector, regardless of the side it came from [11].

## (f) Altitude with maximal wind support and used flight altitude

Birds can vary their flight altitude to optimally use the wind conditions along the route [9,39]. We calculated the altitude at which

each male would have experienced the highest mean wind support, as follows. We computed average mean wind support for 1 h flight intervals for all six altitudes (i.e. allowing a change to any altitude each hour). The altitude with maximal wind support (hereafter 'maxWs' altitude) was then defined as the altitude with highest mean wind support for each 1 h interval.

To estimate the altitude at which males were most likely to have flown, we fitted linear mixed-effect models with the birds' ground speed as the dependent variable and with wind support and cross wind at a given altitude as the explanatory variables. Thus, we fitted seven models: one for each fixed altitude and one for the maxWs altitude as defined above. We included track ID as random effect and used a moving-average correlation structure to control for temporal autocorrelation. We identified the altitude that best explained the observed ground speeds based on a maximum-likelihood approach using Akaike's information criterion (AIC), selecting the model with the lowest AIC. We then compared the mean wind support during the first 50 km at 10 m with that at the most likely flight altitude and assessed whether wind support at both altitudes was correlated (i.e. whether individuals could predict the wind conditions at flight altitude based on the wind close to ground level).

## (g) Wind and flight performance

To assess the influence of the wind on ground and air speed, we used the model as described above and a similar model with air speed (calculated as ground speed minus wind support) as the dependent variable.

For nomadic movements, we expect that individuals will opportunistically cover longer distances over sea if they experience stronger wind support, assuming everything else (e.g. condition) is equal. This implies that the amount of wind support will influence in which part of the breeding range a male pectoral sandpiper will arrive to search for suitable breeding habitat and potentially establish a territory. To test whether males covered longer distances over sea before reaching land with higher wind support, we fitted a linear model with track length as the dependent variable and with mean wind support during the first half of the track as the explanatory variable.

## (h) Wind and direction and timing of departure

To test whether local wind conditions predict the initial flight direction and whether the initial flight direction in turn predicts where the birds reach land (i.e. suitable breeding habitat), we fitted two linear mixed-effect models. In the first model, we used the initial flight direction (mean of first 50 km) as the dependent variable, and wind (i.e. interaction between mean wind direction and wind speed during the first 50 km of flight) as the explanatory variable. In the second model, the final flight direction (i.e. the direction between departure and arrival location) was the dependent variable and the initial flight direction the explanatory variable. In both models, we included date (night in a given year) as a random effect.

Second, we asked whether the expected wind support predicted how many males flew either to the Russian or to the North American Arctic on a given day. Using all observed, directed flights, either to the Russian or to the North American Arctic ($n = 78$), we computed wind support at the mean departure time (22.30 Alaska Daylight Time, AKDT) for every day during which departures took place (27 May–23 June). For each departure night, we then calculated the difference in wind support an individual would have experienced when flying east or west (delta wind support: $\Delta_{W-E}$) and linked this to the actual departures in each direction. We then tested whether the observed wind support was higher than the wind support expected by chance. Using all flights ($n = 85$), we simulated 10 000 random departures for each track within the first and last departure date of each season and compared the mean wind support of the simulated flights with the actual observed mean wind support.

Even if males flew with wind support, this does not necessarily imply that they simply went 'with the flow'. The same pattern could arise if individuals waited for favourable wind conditions to depart to their targeted area. Thus, we tested whether males shifted their departure time to match favourable wind conditions, both within a given day (night) and between days. We calculated wind support at the departure location assuming a direct flight to their actual destination (i.e. assuming that each individual had that location as their goal) at the actual departure times ±12 h in hourly steps (within-day decisions, based on local wind data at ground level) and at the actual departure date ±5 days in daily steps (between-day decisions, based on the reanalysis model at six fixed altitudes). If males indeed timed their departure in response to favourable winds, we expect worse wind conditions (i.e. less wind support) to fly to the same destination in the hours or days before or after the actual departure to that destination.

## (i) Comparing wind support for the shortest route and the actual track

For each altitude, we compared the mean wind support on the actual route with the mean wind support the bird would have experienced had it taken the shortest route. To estimate wind support on the shortest path, we simulated for each individual its movement from the departure to the arrival location, assuming a constant heading towards the latter. We then estimated movement speed using the local wind support and crosswind and the ground speed predicted for such wind conditions. The latter was calculated based on the linear mixed-effect model with observed ground speed as the dependent variable (see §2(f)). To match the temporal pattern of the observed tracks, we assumed that the bird flew 15 min with this predicted ground speed, resulting in the next position on the track. If at any position along the shortest track the predicted ground speed was negative, we conservatively assumed that the bird did not move in this 15 min interval (instead of going backwards).

## (j) Data analysis

We performed all analyses with R [40]. Spatial data were transformed to a polar Lambert azimuthal equal-area projection with longitude origin at Utqiaġvik (156.65° W).

We created the R package 'windR' (available at https://github.com/mpio-be/windR), which provides a set of tools to connect flight tracks with wind data and to calculate the wind support and crosswinds. Additionally, the package provides a set of graphical methods to create particle flow animations.

We fitted linear mixed-effect models with the package 'lme4' [41] and 'nlme' [42]. For multiple predictor models, we used the package 'multcomp' [43] to compute adjusted $p$-values from the corresponding $t$ or $z$ multivariate distribution to account for the correlations between the parameter estimates.

# 3. Results

## (a) Flight description

Males departed between 30 May and 17 June (mean ± s.d.: 7 June ± 3 days, excluding one outlier on 23 June), with a mean local departure time of 22.30 (±3.3 h; range: 13.20–08.20 AKDT). Males flew 7–55 h non-stop over sea (mean ± s.d.: 24 ± 8 h), covering 584–2609 km (mean ± s.d.: 1360 ± 456 km). Thus, the average ground speed was $16 \pm 3 \text{ m s}^{-1}$

$(58 \pm 11\ \text{km h}^{-1})$; the fastest male reached an average speed of $22\ \text{m s}^{-1}$ ($78\ \text{km h}^{-1}$).

The percentage detour was overall low (median: 7%), but ranged between less than 1% and 91%. We classified flights into 'directed flights' (92%, median straightness: 1.1, range: 1.0–2.3) and 'loop flights' (8%, median straightness: 7.8, range: 5.1–11.1; figure 1; electronic supplementary material, tracks). Directed flights either went east (i.e. visited areas in the North American Arctic, 44% in 2012 and 76% in 2014) or west (i.e. to the Russian Arctic, 56% in 2012 and 24% in 2014).

## (b) Wind conditions and wind support during the flights

Wind conditions varied between years (electronic supplementary material, figure S2). In 2012, strong winds ($6$–$10\ \text{m s}^{-1}$) blowing in northwesterly direction dominated, whereas in 2014, wind direction was more variable with lower speeds. Wind speeds over the ocean were on average $1.2\ \text{m s}^{-1}$ higher than over the land (electronic supplementary material, figure S2).

Wind support did not differ much between the six altitudes (maximum difference less than $1.5\ \text{m s}^{-1}$; electronic supplementary material, figure S3). However, if males would have flown at the altitude with maximum wind support during each hour along the track (see Material and methods), they would have experienced $2.2$–$3.4\ \text{m s}^{-1}$ higher wind support in comparison to flying at a fixed altitude with $0.6$–$1.8\ \text{m s}^{-1}$ wind support. The altitude with the highest wind support varied within and between tracks. For all tracks combined, the mean percentage of the track with maximal wind support at a given altitude was 7% at 10 m, 27% at 100 m, 18% at 750 m, 4% at 1500 m, 4% at 2250 m and 39% at 3000 m.

The estimated wind support and crosswind at 750 m fitted best to the observed ground speeds (electronic supplementary material, table S1) and was therefore probably the most used altitude. Thus, all further results are from analyses assuming flights at 750 m (conclusions remain the same for the other six altitudes, details not shown). During the first 50 km of each flight, the mean wind support at 10 m and at 750 m correlated strongly (Pearson's $r = 0.92$, $n = 85$, $p < 0.001$).

## (c) Influence of wind conditions on flight performance

With increasing wind support, ground speed increased, while estimated airspeed (the birds' flying effort) decreased (electronic supplementary material, table S2). However, the effect on ground and estimated air speed decreased with increasing crosswinds (electronic supplementary material, table S2).

Wind support during the first half of the track predicted total track length (electronic supplementary material, table S3), after controlling for direction (i.e. east, west or loop flight). The latter is necessary because land is typically reached earlier when going east. Each additional $1\ \text{m s}^{-1}$ wind support increased the overall track on average by 22 km.

## (d) Influence of wind conditions on the trajectory and on the direction and timing of departure

Overall, 70 out of 85 males (82%) left with a positive wind support at departure (mean wind support during the first 50 km of the track: $4.9\ \text{m s}^{-1}$, range: $0.1$–$14.9\ \text{m s}^{-1}$; figure 1).

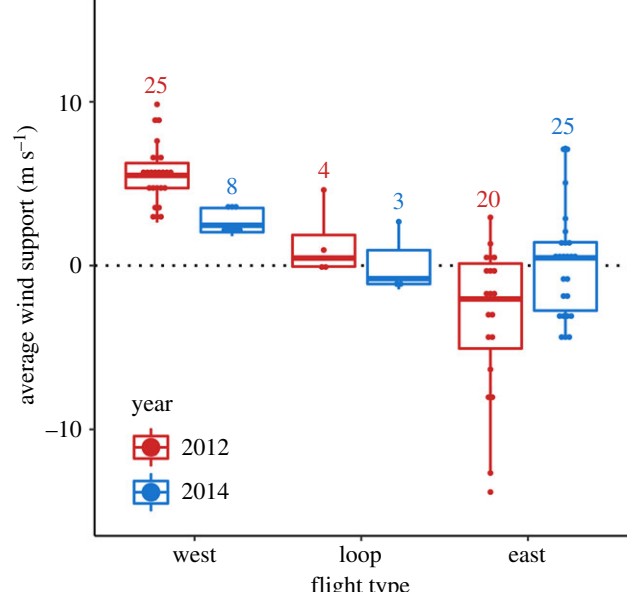

**Figure 2.** Average wind support (calculated based on wind data at approximately 750 m altitude; see Material and methods) for the entire flight in relation to flight type and year. Loop flights are defined as tracks with a low straightness (median = 7.8). Westward and eastward flights are defined as tracks directed toward their respective destination with high straightness (see Material and methods). Shown are box-plots with median (centre line), 25–75th percentile (limits), minimum and maximum values without outliers (whiskers), and outliers (dots). The number above the box indicates the number of tracks in each category. (Online version in colour.)

The other 18% of males left with headwinds ranging from $-0.2$ to $-10.9\ \text{m s}^{-1}$ (mean: $-3.3\ \text{m s}^{-1}$; figure 1). All males that flew west experienced on average positive wind support, whereas males that went east more often faced headwinds (figures 1 and 2; electronic supplementary material, figure S4 and movie available at https://www.youtube.com/watch?v=A-Q5J1wRBUA&feature=youtu.be). The observed mean wind support during the first 50 km of the track was significantly higher than the mean wind support of a random departure within the period of departures in the same season in 97% of cases (based on 10 000 simulations; mean difference: $2.5\ \text{m s}^{-1}$, range: $0.5$–$4.2\ \text{m s}^{-1}$). This was also true when the mean wind support during the entire track was considered, yet the difference between the observed and randomized tracks was smaller (mean difference: $1.7\ \text{m s}^{-1}$, range: $0.2$–$3.1\ \text{m s}^{-1}$; significant for 94% of 10 000 simulations).

The initial flight direction was related to wind direction, with the effect becoming stronger with increasing wind speed (electronic supplementary material, figure S5a and table S4), and was a good predictor of where males became resident next (electronic supplementary material, figure S5b and table S5). As a result, wind conditions at the departure location influenced the number of males that became resident in the Russian versus the North American Arctic (figure 3; electronic supplementary material, table S6).

Whereas initially most birds left with the wind, wind support gradually decreased along the track to average values around zero (electronic supplementary material, figure S6a and table S7). In general, the relationship between wind direction and flight direction (i.e. relative wind support) increased with wind speed (electronic supplementary material, figure S6b).

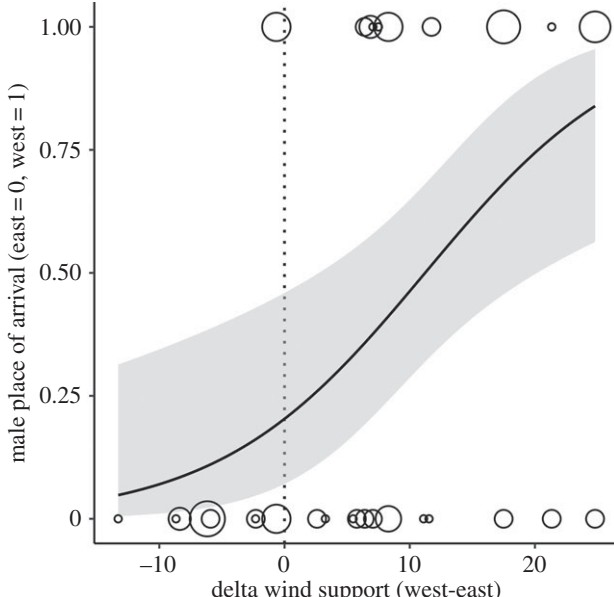

**Figure 3.** Site of arrival in relation to the average difference in wind support for going west (i.e. into the Russian Arctic, $n = 33$) versus east (i.e. into the North American Arctic, $n = 45$) during the first 50 km. Delta wind support was calculated by subtracting the mean wind support for all tracks that went east from the mean wind support for all tracks that went west. Dot size reflects the number of individuals that left during a given night in the same direction ($n = 1–8$). Shown are model estimates (lines) and 95% confidence intervals (grey areas). See electronic supplementary material table S6 for model descriptions.

The difference in wind patterns between years (see above) coincided with differences in the proportion of individuals that flew to the Russian and North American Arctic. In 2012, 56% of the males went west, benefiting from strong tailwinds ($5.4 \pm 4.7$ m s$^{-1}$), while males that went east on average faced headwinds ($-3.0 \pm 5.5$ m s$^{-1}$; figures 1 and 3). In 2014, when winds were generally weaker, only 24% of the males went west and males leaving in both directions on average departed with tailwinds (west: $2.7 \pm 1.6$ m s$^{-1}$; east: $-0.3 \pm 3.7$ m s$^{-1}$; figures 1 and 3; electronic supplementary material, table S8).

Of all males, 35% started a long over-sea flight within 3 days after capture, 57% within 5 days and the remaining males left 6–24 days after capture. Consequently, most males could have adjusted their departure date to use favourable winds to fly towards a hypothetical destination. However, we found no evidence that they would have experienced worse (or better) wind support when shifting their departure by 1–5 nights or by 1–12 h within a night (electronic supplementary material, table S9).

### (e) Wind support on the shortest versus the actual route

Wind support on the actual track was higher than on the shortest route for males going east (mean difference between actual and shortest: $2.1$ m s$^{-1} \pm 0.8$ s.e., $p = 0.019$; electronic supplementary material, table S10) and non-significantly smaller for males going west (mean: $-1.5$ m s$^{-1} \pm 0.9$ s.e., $p = 0.20$). However, males that would have taken the direct route would on average have spent less time flying (assuming they flew with a ground speed predicted for the local wind support and crosswind; see Material and methods), independent of whether they went east (mean difference: $2.57$ h $\pm$ $1.02$ s.e., $p = 0.034$) or west (mean difference: $0.53$ h $\pm$ $1.22$ s.e., $p = 0.96$; electronic supplementary material, table S10).

For loop flights, wind support on the actual track was much higher than it would have been on the direct route (mean difference between actual and shortest: $3.5$ m s$^{-1} \pm 1.3$ s.e., $p = 0.020$). Nevertheless, males spent much more time flying than if they would have taken the shortest route (mean difference: $23.28$ h $\pm 2.43$ s.e., $p < 0.001$). Visual inspection shows that four individuals (114 272, 114 273, 114 279 and 114 307, see electronic supplementary material, tracks) flew a major detour before ending up in the Canadian Arctic.

## 4. Discussion

Little is known about the mechanisms underlying nomadic movements [44]. In this study, we focused on nomadic movements by a polygynous shorebird during the breeding season [27]. Our results suggest that the direction in which pectoral sandpipers flew to sample other potential breeding sites is influenced by the strength and direction of the winds at the departure location. We show that most males flew with wind support, especially those flying to the Russian Arctic. We found no evidence that males waited for optimal wind to leave to a specific target area. Wind conditions influenced both in which direction males left and how far they flew and thus ultimately in which region they arrived and potentially reproduced. Between-season variation in wind conditions influenced the proportion of males that sampled sites in the Russian versus the North American Arctic. For six males with two recorded long flights within the same season, three left twice in the same direction, while the other three flew in opposite directions (electronic supplementary material, tracks). In conclusion, most male pectoral sandpipers seem to opportunistically fly in the direction that provided wind support. As expected, higher wind support led to higher ground speeds at lower air speeds and to longer total flight distances.

### (a) Flight characteristics and wind support

The choice of the most favourable altitude is an important part of an individual's adaptive exploitation of winds [2,3,9]. Our results suggest that males predominantly flew at approximately 750 m.a.s.l., which falls within the range estimated by radar within the Beringia region [38]. We found no evidence that birds adjusted their altitude during flight to obtain maximal wind support (electronic supplementary material, table S1). However, changes in flight altitude did not strongly change wind support (electronic supplementary material, figure S3). Only further studies using transmitters that measure atmospheric pressure will be able to reveal the actual flight altitude.

The majority of males left our study site by flying over the ice-covered sea, rather than over the land where they could have rested or assessed local breeding conditions continuously. Flying over the sea was typically a shorter route to their destination. Higher wind speeds over the sea would have benefited individuals that flew with tailwind, but males facing headwinds would have done better by flying over the land.

### (b) Influence of wind conditions on flight performance

Males adjusted their air speed depending on experienced wind support, flying faster with headwinds and slower

with tailwinds (electronic supplementary material, table S2). This relationship is expected as a way to minimize energy expenditure [34,35] and generally described for birds [45]. Ground speeds increased with tailwinds and decreased with headwinds, but these effects were weakened by cross winds, presumably because individuals changed their heading to compensate for wind drift. Overall, our results suggest that wind support likely had a substantial influence on the energy expenditure of males. With tailwinds, males flew slower, but moved faster and further than males flying with headwinds (electronic supplementary material, tables S2 and S3). On one occasion, a male turned around, presumably to avoid an approaching storm with strong headwinds (see electronic supplementary material, movie available at https://www.youtube.com/watch?v=A-Q5J1wRBUA&feature= youtu.be, 10 June 2014).

## (c) Influence of wind conditions on breeding site sampling

The males' initial flight direction was influenced by wind direction and speed, such that most birds initially left with tailwinds, especially when wind speeds were high (figures 2*a,b* and 3). The wind conditions at ground level (10 m) correlated strongly with those at the most likely flight altitude (750 m). Males could thus use the wind conditions on the ground to predict the wind conditions they would experience during their flight. For all flights, the observed wind support was higher than the wind support individuals would have experienced had they left on a random day within the same period. However, we found no evidence that males waited for favourable wind conditions to depart (electronic supplementary material, table S9). Given the short breeding season and the unpredictable conditions (opportunities to mate) at other potential breeding sites, waiting for optimal winds to reach a particular goal may reduce the probability to reproduce in a given year.

Along the track, the relationship between wind direction and track direction weakened. Thus, males initially moved 'with the flow', but most kept a rather constant flight direction, which usually resulted in a decrease of wind support along the track. Some individuals initially flew in a northerly direction, before turning east or west (figure 1). These movements mirror the strategy of 'adaptive drift' [46], where birds can gain time and energy by allowing an initial ground speed increase due to drift followed by displacement compensation toward the end of the trip. Given the shape of the breeding range this compensation was necessary to reach land (figure 1). Consequently, most males did not simply fly 'with the flow' for the entire track over sea, but may have aimed towards a broad goal. Once the males reached land, the majority settled after a short distance, but a few continued flying longer distances. The factors that determine the local site selection remain unknown, but direct cues about the suitability of a potential breeding site such as snow cover or presence of conspecifics may be more important than wind conditions.

Several males faced moderate to strong headwinds directly after departure and over their entire journey (figures 1 and 3). This suggests that some individuals may have a preferred direction. Only 2% of males returned to our study area between years and these were typically males that successfully sired offspring [31]. Thus, males may base their movement decisions on experience during past breeding seasons.

Strikingly, however, males never flew west towards Russia with headwinds (figures 1 and 3). This corroborates a study at the Taimyr Peninsula, which reported that after 7 years without breeding records, nests were found in a year with easterly winds [47]. We speculate that this pattern is related to the autumn migration route. Pectoral sandpipers predominantly winter in South America [48] and all males that went to the Russian Arctic flew back east to Alaska and along the Canadian coast towards the Hudson Bay [27] (B.K. 2015–2019, unpublished data). Consequently, males that fly west into the Russian Arctic will later need more resources to cover a much longer migration distance, whereas males flying east into the Canadian Arctic were already moving in the direction of their later destination.

Assuming that time is an important resource [27,31], the observed 'loop' flights with long detours remain puzzling. We briefly discuss potential explanations. (i) Males 'overshoot' land during migration. Although a common phenomenon in bird migration [28], it seems unlikely because males already arrived (and started to compete) at a breeding site. (ii) Some, possibly naive, individuals use the wind to explore new breeding sites, but when failing to find land, turn around. Pectoral sandpipers are common vagrants [49] and hence males may have a tendency to explore. (iii) 'Loop' flights may be an outcome of group behaviour. Males typically fly in small groups (approx. 10–100 individuals, our personal observations, see also [28]). If only few individuals have a goal (i.e. have decided to return to a previous breeding area, see above), these individuals may act as 'leaders', pulling a group in a particular direction. However, if a group contains a mix of individuals that want to fly east and west, the group might initially head in an intermediate direction, i.e. northwards, but later split when the directional conflict becomes too large [50,51].

## 5. Conclusion

Our study shows that wind conditions influence the direction and the speed of flights of male pectoral sandpipers during the breeding season. Males seem to be highly flexible in where they go and the prevailing wind conditions can explain both within- and between-year variation in the distribution of males across their arctic breeding range. The importance of collective behaviour (group decisions) and the potential role of a small number of 'leaders' that decide to return to a previous breeding site remains to be studied. The observed patterns could also be driven by males following females, which themselves sample multiple breeding sites (B.K. 2018-2019, unpublished data) and were present at the time when males made the observed flights. Variability in wind patterns over the Arctic Ocean in combination with unpredictable breeding site quality could favour nomadic movements in a species with no mate fidelity and female-only parental care, or at least will not create the environmental prerequisites for wind-optimized flyways of fixed migration routes as found for example in the Northern Hemisphere [52].

The results of this study illustrate that for nomadic species—or more generally for species that are not site faithful—decisions about where to go could be influenced by the movement of the medium they travel through (wind in the case of birds), such that the costs of the movement (flight costs) can be reduced. If the distribution of the aimed for

resources (e.g. safety from predators, food, mates) is unpredictable over a large spatial scale, individuals may benefit from using the energetically cheapest routes to travel to a potentially suitable site. When considering 'optimal migration theory', these movements initially mirror 'adaptive drift' [2], with the difference that nomadic animals only need to compensate such that they reach any potential site (e.g. pectoral sandpipers need to reach land when flying over the sea ice), whereas highly philopatric species will have to compensate drift completely to reach their particular goal. In this context, comparing the costs linked to movements to different parts of the species's (breeding) range can be highly valuable, because it can explain seasonal or between-year variation in local (breeding) density. This also implies that in species with low levels of philopatry, local fluctuations in numbers cannot be used to estimate population size. Our study also implies that estimating changes in population size based on local counts in a species with low levels of philopatry requires prior knowledge about the mechanisms underlying the distribution of the species across its range.

Data accessibility. The datasets generated and analysed for this study, including code used for statistical analysis and figure production, are available at https://osf.io/amd3r/.

Authors' contributions. B.K. and M.V. initiated the study; J.K. analysed the data with input from B.K. and M.V; all authors interpreted the results; J.K. and B.K. wrote the paper with input from M.V.

Competing interests. We declare we have no competing interest.

Funding. This study was funded by the Max Planck Society (to B.K.). J.K. was supported by the International Max Planck Research School for Organismal Biology.

Acknowledgements. We thank the members of the Kempenaers Department and in particular Wolfgang Forstmeier for discussion and suggestions and Jelle Loonstra and one anonymous referee for constructive comments.

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
