## [Reviewer comments · Proceedings of the Royal Society B: Biological Sciences]

Review History

RSPB-2019-1119.R0 (Original submission)

Review form: Reviewer 1

Recommendation

Major revision is needed (please make suggestions in comments)

Scientific importance: Is the manuscript an original and important contribution to its field?

Acceptable

General interest: Is the paper of sufficient general interest?

Acceptable

Quality of the paper: Is the overall quality of the paper suitable?

Acceptable

Is the length of the paper justified?

Yes

Should the paper be seen by a specialist statistical reviewer?

No

Do you have any concerns about statistical analyses in this paper? If so, please specify them explicitly in your report.

Yes

It is a condition of publication that authors make their supporting data, code and materials available - either as supplementary material or hosted in an external repository. Please rate, if applicable, the supporting data on the following criteria.

Is it accessible?

N/A

Is it clear?

N/A

Is it adequate?

N/A

Do you have any ethical concerns with this paper?

No

Comments to the Author

Please see my comments in the attached file. (See Appendix A)

Review form: Reviewer 2

Recommendation

Major revision is needed (please make suggestions in comments)

Scientific importance: Is the manuscript an original and important contribution to its field?

Good

General interest: Is the paper of sufficient general interest?

Good

Quality of the paper: Is the overall quality of the paper suitable?

Marginal

Is the length of the paper justified?

Yes

Should the paper be seen by a specialist statistical reviewer?

No

Do you have any concerns about statistical analyses in this paper? If so, please specify them explicitly in your report.

Yes

It is a condition of publication that authors make their supporting data, code and materials available - either as supplementary material or hosted in an external repository. Please rate, if applicable, the supporting data on the following criteria.

Is it accessible?

Yes

Is it clear?

Yes

Is it adequate?

Yes

Do you have any ethical concerns with this paper?

No

Comments to the Author

General Comments:

The authors investigated how experienced wind conditions influenced the breeding season movements in a “nomadic” polygynous shorebird. To answer this question, the authors used a subset of previous published migratory tracks of male pectoral sandpipers (hereafter: PeSa) that were tagged in Barrow, subsequently the authors linked the breeding season movements with the wind-conditions en route and at departure. While I really like the interesting observation and patterns between the different “groups” of migrants (East, West & Loop; Fig. 3), I’m sorry to say that I dislike the current framing of the MS. The main reason for this is the lack of causality and the fact that the authors over-emphasize the effect of wind on breeding season movements (see for instance the large contradiction between lines 388-389 and lines 358-359). Based on the provided data and analysis I’m currently not convinced that the authors provide a solid explanation of the “nomadic” breeding season movements of PeSas (totally wind-driven).

Therefore I suggest that authors either perform some additional analysis (see comments below) or change the framing of the MS. For instance, the authors could frame the paper more descriptive/observational and discuss that some movements are only made with enough wind-assistance while others (to the east) are also performed without a strong wind-assistance. The authors could then speculate whether this is caused by potential individual differences; sired offspring at a leek, age, group-composition and wind-conditions.

Also, as the authors aim to understand the general “breeding season movements of PeSas” they should be very careful when excluding certain tracks. With the current sub-setting of tracks, especially rule (iii) in line 105 and the removal of overland tracks in lines 121-122 the authors remove part of the natural existing variation in “breeding season movements of PeSas”. Overland tracks are also part of the breeding season movements and by excluding these tracks you can no longer talk about breeding season movements in general, but only about breeding season movements over sea-ice.

Major Comments:

Lines 56-57: Argument behind the comma is not logically true; there could also be variable goals (i.e. nomadism) together with goal-oriented flights, these aren’t opposites as presented here. Especially when individuals have knowledge about the area to the west and the area to the east they can flexibly adjust their goal to this. That would be goal-oriented nomadism! Thus I would suggest to re-formulate this last part of an overall well-written introduction.

Lines 61-68. The authors give a very nice description on the ecology of MALE PeSas. However, as the authors mention in line 64-65 these “horny” PeSas males are in search of mating opportunities. This makes the idea of nomadism i.e. “going with the flow” less likely and “going with the females” much more (see also your earlier publication on this idea of “mate-searching” [27]). Consequently it is of great importance to give at least some basic information on the migration ecology of FEMALE PeSas. For instance, is there anything known about their timing of migration in Barrow? Do they arrive later, earlier or at the same time? And could it be that the females are actually the nomadic individuals, while the males just follow the females?

Lines 68-70: True, however this requires sampling the departure decision of individuals multiple times. Since you only have limited repeated measurements of the same individual, it is difficult for you to make generalizations. Especially because one half of the repeated measured individuals does the opposite of the other half you cannot really come to a conclusion on this. Please reframe the entire manuscript accordingly.

Lines 77-78: In my opinion you can only answer this question with repeated tracks of the same

individual or by using a more mechanistic modelling approach like Revell & Somveille 2017 Sci. Rep. 7: 9870. With the mechanistic modelling approach you could include all the possible available “goals” when you combine the observed goals with remote sensing and GIS technologies to identify breeding/leaking areas. Then, by combining the wind-conditions and possible goals you could perform a simulation exercise whereby you have 1-million particles leaving from Barrow at a certain moment. By varying the importance of the different components in your model you can then infer which component is the most important and explains the breeding season movements of male PeSas.

Lines 87-89: Again, you can describe the relationship between wind and your observation (which you do very neatly), but you cannot assess whether your observation depends on this. You should stick to your first aim and discuss what this might mean, but you shouldn't aim to do something you currently cannot.

Line 105, point (iii): This limits the generalizations you can make about the relation between your observed tracks and wind conditions to those birds that departed over sea. This limitation should be acknowledged in your discussion as it might change the relation you find. You could also add a statement or analysis that choosing for this subset of individuals does not influence your results.

Line 120 (point ii): Same as above, by excluding this data you limit how general your results are. In this case you analyze track lengths that are shorter than actually observed and your found relationship with wind conditions is only valid for this part of the track. I don't mind this, your analysis is still informative, but you must acknowledge the consequences of excluding this data. Best would be to see an analysis with and without the data excluded. That would give additional information about whether flights over sea are in fact more or less dependent on wind than flights over land (with additional cues).

Lines 193-194: True, but you show that they flew both with a tailwind and a headwind. And your additional analysis (lines 195 - 201) shows that they didn't wait for favorable winds, instead you might interpret your results as if they flew with a headwind on purpose (as they didn't wait for a tailwind).

Lines 228-230: See above, this is only true for those parts of the flight that were over the sea. It might be good to acknowledge that.

Lines 245-247: “Optimal” is not the same as the altitude with maximum wind support. I would suggest referring to it as the altitude with maximum wind support as there is no information whether this is also optimal or not. (see also lines: 325-327)

Lines 252-254: Please supply a statistic that evaluates the Goodness-of-fit of this model.

Lines 267-272: Is this a significant proportional difference between wind conditions for the different groups?

Line 305-306: They might also have expended more energy by flying for a longer time, therefore please quantify this, acknowledge this or remove this slightly misleading statement.

Line 309: “seemingly” nomadic is not necessarily the same as a large variation in routes.

Line 352: A constant flight direction makes a goal-oriented flight more likely especially as your results show not all of them “go with the flow” .

Minor Comments:

Line 73: “how individual pectoral sandpipers respond to wind conditions” □ Where? At the breeding grounds? En route? At departure? At arrival?

Line 79: “centre of the range” □ “centre of their distribution range”

Lines 97-99: The authors should mention the continuous duty-cycle of the tags and the number of received Argos-locations per hour.

Line 112: I would suggest to explicitly mention that these are repeated WITHIN-season tracks and not between-seasons.

Lines 131: ~80km: I would provide this information in degrees, the ERA-Interim grid is also in degrees.

Lines 138-140: According to your results, PeSas fly at an altitude of 750m, thus if they use local wind-conditions for their departure-decision you should at least show that there is no difference between ground-level wind-conditions at departure and wind-conditions at 750m.

Line 164: “a change” does this mean 1 change in altitude per hour? Also, do you allow changes between 0 □ 3000m or just changes between consecutive altitudes?

Lines 329-331: This could be, but the "optimal" temperature zone will be species specific. I would suggest remove this part of speculation and argue that the authors need to measure (e.g. GPS-transmitters) the actual flight-altitude before discussing the flight-altitude.

Lines 387-396: After reading the MS I was wondering whether the authors are able to infer the status (alive/dead) of a tagged PeSa (e.g. Sergio et al. 2018) and if so, whether the authors did observe a proportional difference in mortality-rate/tag-los between the different movements (west, east and loop or amount of wind-assistance). Also, and just out of curiosity, I was wondering whether the authors did clip a small part of the feathers (see also Yohannes et al. 2012 Ibis). It would be very interesting to see whether the breeding season movements differ between the two different age-classes and whether the interaction with wind differs per age-class.

Decision letter (RSPB-2019-1119.R0)

09-Jul-2019

Dear Professor Kempenaers:

We are writing to inform you that your manuscript RSPB-2019-1119 entitled "Wind conditions influence breeding season movements in a nomadic polygynous shorebird" has, in its current form, been rejected for publication in Proceedings B.

This action has been taken on the advice of referees, who have recommended that substantial revisions are necessary. With this in mind we would be willing to consider a resubmission, provided the comments of the referees and the Associate Editor are fully addressed. It is important to note that this is not a provisional acceptance.

The resubmission will be treated as a new manuscript, however, we will approach the same reviewers if they are available and it is deemed appropriate to do so by the Editor. Please note that resubmissions must be submitted within six months of the date of this email. In exceptional circumstances, extensions may be possible if agreed with the Editorial Office. Manuscripts submitted after this date will be automatically rejected.

Sincerely,

Proceedings B
 mailto: proceedingsb@royalsociety.org

=====

Associate Editor, Comments to Author:

Two referees reviewed your manuscript and both express concerns about the lack of statistical support for the claim that flights are opportunistic and driven by wind conditions (ln. 313-315 and 388-389). They also highlight several other important issues that you need to address; notably the role of distribution of females and nests in driving male flight preferences. You need to carefully address all reviewers' concerns in order to demonstrate that your manuscript is novel enough to be published in Proceedings B.

====

Reviewers' Comments to Author:

Referee: 1

Please see my comments in the attached file.

===

Referee: 2

General Comments:

The authors investigated how experienced wind conditions influenced the breeding season movements in a "nomadic" polygynous shorebird. To answer this question, the authors used a subset of previous published migratory tracks of male pectoral sandpipers (hereafter: PeSa) that were tagged in Barrow, subsequently the authors linked the breeding season movements with the wind-conditions en route and at departure. While I really like the interesting observation and patterns between the different "groups" of migrants (East, West & Loop; Fig. 3), I'm sorry to say that I dislike the current framing of the MS. The main reason for this is the lack of causality and the fact that the authors over-emphasize the effect of wind on breeding season movements (see for instance the large contradiction between lines 388-389 and lines 358-359). Based on the provided data and analysis I'm currently not convinced that the authors provide a solid explanation of the "nomadic" breeding season movements of PeSas (totally wind-driven). Therefore I suggest that authors either perform some additional analysis (see comments below) or change the framing of the MS. For instance, the authors could frame the paper more descriptive/observational and discuss that some movements are only made with enough wind-assistance while others (to the east) are also performed without a strong wind-assistance. The authors could then speculate whether this is caused by potential individual differences; sired offspring at a leek, age, group-composition and wind-conditions. Also, as the authors aim to understand the general "breeding season movements of PeSas" they should be very careful when excluding certain tracks. With the current sub-setting of tracks, especially rule (iii) in line 105 and the removal of overland tracks in lines 121-122 the authors remove part of the natural existing variation in "breeding season movements of PeSas". Overland tracks are also part of the breeding season movements and by excluding these tracks you can no longer talk about breeding season movements in general, but only about breeding season movements over sea-ice.

Major Comments:

Lines 56-57: Argument behind the comma is not logically true; there could also be variable goals (i.e. nomadism) together with goal-oriented flights, these aren't opposites as presented here. Especially when individuals have knowledge about the area to the west and the area to the east they can flexibly adjust their goal to this. That would be goal-oriented nomadism! Thus I would suggest to re-formulate this last part of an overall well-written introduction.

Lines 61-68. The authors give a very nice description on the ecology of MALE PeSas. However, as the authors mention in line 64-65 these "horny" PeSas males are in search of mating opportunities. This makes the idea of nomadism i.e. "going with the flow" less likely and "going with the females" much more (see also your earlier publication on this idea of "mate-searching"

[27]). Consequently it is of great importance to give at least some basic information on the migration ecology of FEMALE PeSas. For instance, is there anything known about their timing of migration in Barrow? Do they arrive later, earlier or at the same time? And could it be that the females are actually the nomadic individuals, while the males just follow the females?

Lines 68-70: True, however this requires sampling the departure decision of individuals multiple times. Since you only have limited repeated measurements of the same individual, it is difficult for you to make generalizations. Especially because one half of the repeated measured individuals does the opposite of the other half you cannot really come to a conclusion on this. Please reframe the entire manuscript accordingly.

Lines 77-78: In my opinion you can only answer this question with repeated tracks of the same individual or by using a more mechanistic modelling approach like Revell & Somveille 2017 Sci. Rep. 7: 9870. With the mechanistic modelling approach you could include all the possible available "goals" when you combine the observed goals with remote sensing and GIS technologies to identify breeding/lecking areas. Then, by combining the wind-conditions and possible goals you could perform a simulation exercise whereby you have 1-million particles leaving from Barrow at a certain moment. By varying the importance of the different components in your model you can then infer which component is the most important and explains the breeding season movements of male PeSas.

Lines 87-89: Again, you can describe the relationship between wind and your observation (which you do very neatly), but you cannot assess whether your observation depends on this. You should stick to your first aim and discuss what this might mean, but you shouldn't aim to do something you currently cannot.

Line 105, point (iii): This limits the generalizations you can make about the relation between your observed tracks and wind conditions to those birds that departed over sea. This limitation should be acknowledged in your discussion as it might change the relation you find. You could also add a statement or analysis that choosing for this subset of individuals does not influence your results.

Line 120 (point ii): Same as above, by excluding this data you limit how general your results are. In this case you analyze track lengths that are shorter than actually observed and your found relationship with wind conditions is only valid for this part of the track. I don't mind this, your analysis is still informative, but you must acknowledge the consequences of excluding this data. Best would be to see an analysis with and without the data excluded. That would give additional information about whether flights over sea are in fact more or less dependent on wind than flights over land (with additional cues).

Lines 193-194: True, but you show that they flew both with a tailwind and a headwind. And your additional analysis (lines 195 - 201) shows that they didn't wait for favorable winds, instead you might interpret your results as if they flew with a headwind on purpose (as they didn't wait for a tailwind).

Lines 228-230: See above, this is only true for those parts of the flight that were over the sea. It might be good to acknowledge that.

Lines 245-247: "Optimal" is not the same as the altitude with maximum wind support. I would suggest referring to it as the altitude with maximum wind support as there is no information whether this is also optimal or not. (see also lines: 325-327)

Lines 252-254: Please supply a statistic that evaluates the Goodness-of-fit of this model.

Lines 267-272: Is this a significant proportional difference between wind conditions for the different groups?

Line 305-306: They might also have expended more energy by flying for a longer time, therefore please quantify this, acknowledge this or remove this slightly misleading statement.

Line 309: "seemingly" nomadic is not necessarily the same as a large variation in routes.

Line 352: A constant flight direction makes a goal-oriented flight more likely especially as your results show not all of them "go with the flow" .

Minor Comments:

Line 73: "how individual pectoral sandpipers respond to wind conditions" □ Where? At the breeding grounds? En route? At departure? At arrival?

Line 79: "centre of the range" □ "centre of their distribution range"

Lines 97-99: The authors should mention the continuous duty-cycle of the tags and the number of received Argos-locations per hour.

Line 112: I would suggest to explicitly mention that these are repeated WITHIN-season tracks and not between-seasons.

Lines 131: ~80km: I would provide this information in degrees, the ERA-Interim grid is also in degrees.

Lines 138-140: According to your results, PeSas fly at an altitude of 750m, thus if they use local wind-conditions for their departure-decision you should at least show that there is no difference between ground-level wind-conditions at departure and wind-conditions at 750m.

Line 164: "a change" does this mean 1 change in altitude per hour? Also, do you allow changes between 0-3000m or just changes between consecutive altitudes?

Lines 329-331: This could be, but the "optimal" temperature zone will be species specific. I would suggest remove this part of speculation and argue that the authors need to measure (e.g. GPS-transmitters) the actual flight-altitude before discussing the flight-altitude.

Lines 387-396: After reading the MS I was wondering whether the authors are able to infer the status (alive/dead) of a tagged PeSa (e.g. Sergio et al. 2018) and if so, whether the authors did observe a proportional difference in mortality-rate/tag-loss between the different movements (west, east and loop or amount of wind-assistance). Also, and just out of curiosity, I was wondering whether the authors did clip a small part of the feathers (see also Yohannes et al. 2012 Ibis). It would be very interesting to see whether the breeding season movements differ between the two different age-classes and whether the interaction with wind differs per age-class.

Author's Response to Decision Letter for (RSPB-2019-1119.R0)

See Appendix B.

RSPB-2019-2789.R0

Review form: Reviewer 1

Recommendation

Accept with minor revision (please list in comments)

Scientific importance: Is the manuscript an original and important contribution to its field?

Good

General interest: Is the paper of sufficient general interest?

Good

Quality of the paper: Is the overall quality of the paper suitable?

Good

Is the length of the paper justified?

Yes

Should the paper be seen by a specialist statistical reviewer?

No

Do you have any concerns about statistical analyses in this paper? If so, please specify them explicitly in your report.

No

It is a condition of publication that authors make their supporting data, code and materials available - either as supplementary material or hosted in an external repository. Please rate, if applicable, the supporting data on the following criteria.

Is it accessible?

No

Is it clear?

N/A

Is it adequate?

N/A

Do you have any ethical concerns with this paper?

No

Comments to the Author

My comments are attached in a separate document. (See Appendix C)

Review form: Reviewer 2 (A.H. Jelle Loonstra)

Recommendation

Accept with minor revision (please list in comments)

Scientific importance: Is the manuscript an original and important contribution to its field?

Good

General interest: Is the paper of sufficient general interest?

Excellent

Quality of the paper: Is the overall quality of the paper suitable?

Excellent

Is the length of the paper justified?

Yes

Should the paper be seen by a specialist statistical reviewer?

No

Do you have any concerns about statistical analyses in this paper? If so, please specify them explicitly in your report.

No

It is a condition of publication that authors make their supporting data, code and materials available - either as supplementary material or hosted in an external repository. Please rate, if applicable, the supporting data on the following criteria.

Is it accessible?

Yes

Is it clear?

Yes

Is it adequate?

Yes

Do you have any ethical concerns with this paper?

No

Comments to the Author

Dear Johannes and co-authors,

I'm happy to see this nicely revised manuscript come back and I would like to thank the authors for addressing all comments. In particular I want to point out that that the addition of a null-model as explained in lines 205-209 makes a great attribution and strengthens the story!

Furthermore, I appreciate the investment you made by revising the text while taking into account the proposed suggestions, as a result I believe that the interpretation of the results has become more fair and that all readers are informed on the beautiful complexity of ecology. Altogether, I would cite this manuscript as evidence that wind conditions influence the breeding season movements of male Pectoral Sandpipers.

My compliment on this cool paper on a cool system which I am sure you will able to finalize in short time!

All the best!

Jelle Loonstra

Minor comments/suggestions

Line 39: Even though I hate it be self-promoting;). You could stress the importance of in-flight wind-conditions by citing our recent paper: Loonstra et al. 2019; Ecol. Lett.

Line 43: Personally I really like the Honey Buzzard story of Vansteelant et al. 2016 J. Anim. Ecol. as a good example of a bird adjusting its route with respect to the most favorable winds.

Line 75: missing "." after [30].

Line 344: Suggestion: "they could potentially reproduce" □ "they arrive and potentially reproduce"

Line 346: To prevent any confusion, it might be good to add: "within the same season" after "long flights".

Line 429: "which themselves sample multiple breeding sites": To make it even more complicated;) But very interesting and cool interesting to have these female tracks!

Lines 430-433: You might want to put this in the perspective of the findings from Kranstauber et al. 2012 Ecol. Lett.

Line 449: Raymond □ R

Line 479: RG □ RHG

Decision letter (RSPB-2019-2789.R0)

20-Dec-2019

Dear Professor Kempenaers:

Your manuscript has now been peer reviewed and the reviews have been assessed by an Associate Editor. The reviewers' comments (not including confidential comments to the Editor) and the comments from the Associate Editor are included at the end of this email for your reference. As you will see, the reviewers and the Editors have raised some concerns with your manuscript and we would like to invite you to revise your manuscript to address them.

Research ethics:

Use of animals and field studies:

Please submit a copy of your revised paper within three weeks. If we do not hear from you within this time your manuscript will be rejected. If you are unable to meet this deadline please let us know as soon as possible, as we may be able to grant a short extension.

Best wishes,
Dr Daniel Costa
mailto:proceedingsb@royalsociety.org

Associate Editor Board Member

Comments to Author:

The two reviewers that evaluated the previous version of your ms are satisfied with your revised manuscript. However, you need to make a better case for the general applicability of your results. At the moment, the discussion is too focused on your model species; you need to address how your results inform optimal migration theory. Referee 2 provides excellent suggestions on how to address this problem.

Reviewer(s)' Comments to Author:

Referee: 2

Comments to the Author(s).

Dear Johannes and co-authors,

I'm happy to see this nicely revised manuscript come back and I would like to thank the authors for addressing all comments. In particular I want to point out that the addition of a null-model as explained in lines 205-209 makes a great contribution and strengthens the story! Furthermore, I appreciate the investment you made by revising the text while taking into account the proposed suggestions, as a result I believe that the interpretation of the results has become more fair and that all readers are informed on the beautiful complexity of ecology. Altogether, I would cite this manuscript as evidence that wind conditions influence the breeding season movements of male Pectoral Sandpipers.

My compliment on this cool paper on a cool system which I am sure you will be able to finalize in short time!

All the best!

Jelle Loonstra

Minor comments/suggestions

Line 39: Even though I hate it be self-promoting;). You could stress the importance of in-flight wind-conditions by citing our recent paper: Loonstra et al. 2019; Ecol. Lett.

Line 43: Personally I really like the Honey Buzzard story of Vansteelant et al. 2016 J. Anim. Ecol. as a good example of a bird adjusting its route with respect to the most favorable winds.

Line 75: missing "." after [30].

Line 344: Suggestion: "they could potentially reproduce" □ "they arrive and potentially reproduce"

Line 346: To prevent any confusion, it might be good to add: "within the same season" after "long flights".

Line 429: "which themselves sample multiple breeding sites": To make it even more complicated;) But very interesting and cool interesting to have these female tracks!

Lines 430-433: You might want to put this in the perspective of the findings from Kranstauber et al. 2012 Ecol. Lett.

Line 449: Raymond □ R

Line 479: RG □ RHG

Referee: 1

Comments to the Author(s).

My comments are attached in a separate document.

Author's Response to Decision Letter for (RSPB-2019-2789.R0)

See Appendix D.

Decision letter (RSPB-2019-2789.R1)

21-Jan-2020

Dear Professor Kempenaers

I am pleased to inform you that your manuscript entitled "Wind conditions influence breeding season movements in a nomadic polygynous shorebird" has been accepted for publication in Proceedings B.

Open Access

Paper charges

Sincerely,

Dr Daniel Costa

Associate Editor:

Board Member

Comments to Author:

I have read your revised version and feel that you have satisfactorily incorporated the comments made by referee 1. In particular, the added paragraph at the end of the conclusions clearly explain why the results of your study have important implications for the general field of avian migration studies.

Appendix A

Review of: Wind conditions influence breeding season movements in a nomadic polygynous shorebird

Proceedings of the Royal Society B

Johannes Krietsch, Mihai Valcu and Bart Kempenaers

In this paper, Krietsch *et al.* explore the relationships between wind patterns and breeding-season movement patterns in pectoral sandpipers. Using tracking data of male sandpipers, they analyze wind speed (and related metrics) along flight paths with the goal of understanding whether males select breeding sites based on wind patterns (vs. having a more directed goal). Overall, I found the analyses well described and the results clearly presented. However, I found the framing (introduction & discussion) of the manuscript difficult to understand at times, and think the paper should either be re-framed or needs alternative analyses to truly test the hypotheses the authors propose.

I agree that the movements of these breeding-season males can be characterized as nomadic because they are irregular. However, I disagree with the authors' statements that movements must be "without a specific geographical target" (line 69) in order to be nomadic. In many cases nomadic movements do appear to be directed (e.g. Pedler *et al.* 2014). In the case given here, nomadic movements could have a known destination *and* use wind support if individuals choose an optimal day to depart. The authors do perform an analysis to test this hypothesis (and reject it), but overall I disagree with the idea that this manuscript tests *whether* sandpipers are nomadic. It was not clear to me whether this issue was primarily conceptual (i.e., a difference in the definition of nomadism) or semantic (i.e., unclear wording).

The authors also aim to test for a mechanism of nomadic movements, in other words whether flights are opportunistic in their direction depending on wind. They rely on conclusions such as "most males flew with wind support (line 312)" to support the idea that flights are opportunistic in their direction. In my opinion, a null model would be a more powerful way to test this hypothesis, since it would really answer the question "do more males fly with wind support than would be expected if they had a known destination"? As it is, I do not see a clear quantitative conclusion; for instance, Fig. 3 shows that a substantial proportion of the population flew with average negative wind support. Given the distribution of destination sites and wind speeds, the distribution of wind support *could* be random. For instance, an appropriate null model could randomize departure dates and destinations for each path, then calculate the average wind support and compare that to what birds actually experienced. (Other null models could also be valid, but I wanted to give an example for clarity.)

In addition, neither the introduction nor discussion gives enough background on the distribution of females and nests. For instance, where are females present? Are males changing their probability of encountering females by choosing some locations vs. others? This seems like a very important aspect of understanding the nomadic movements, since the benefit of reaching a destination is finding a mate. In the introduction, the authors state that "the costs of reaching a specific site can become part of the quality characteristics of this site." (Line 71). This is an interesting and straightforward way of thinking about travel costs in movement analyses! However, without other information (i.e., site quality upon arrival), it is hard to draw any conclusion about whether traveling with the wind is actually beneficial. To answer these

questions, it would be very helpful to include information on the locations of females, nest success, or other metrics of site quality upon arrival.

As an alternative to these analyses, I think this paper already provides a clear description of the natural history of this system and the flights of pectoral sandpipers. The fact that flights tend to have wind support and that inter-annual differences in flight destinations depend on wind appears to be a new result. In this case, I would think the paper would be more suited to an more specific (e.g. ornithology) journal.

I have included a few more detailed comments below.

Introduction:

- Line 47: I wouldn't necessarily characterize this as "higher plasticity," since it is plastic in space but not time.
- Line 50: "individuals might face trade-offs with optimal timing." What does this mean? Tradeoffs between what and what?
- Lines 43-57: I struggled somewhat to grasp the goal of this paragraph. It seems to aim to bridge the gap between the first paragraph about optimal migration and the third, about nomadic movements, but I'm not sure exactly where the link is. It seems unnecessary as written, so I would suggest removing it or editing it to make the goals more clear.
- Paragraph lines 79-86: This comes a little out of the blue. Can you introduce these concepts of performance as they relate to wind and what the alternatives might be? This section is not my area of expertise, but I am having trouble understanding how it could be increasing wind support *wouldn't* result in lower costs.

Methods

- Line 128: I think this "sinuosity" would be more accurately called "straightness" or "efficiency," since you are not measuring changes in direction or step lengths (Benhamou 2004)
- Movement tracks have data points every 15 minutes, whereas wind data is every 6 hours. Are the results sensitive to the time scale of the movement tracks? This could be important, especially if wind speeds are variable enough within each 6-hour window. I would have liked to see a sensitivity analysis of the effect of coarsening the temporal resolution of the movement data.
- Section (g): I need more explanation of this hypothesis! Why would this be true only if individuals are nomadic? I am also not sure how the linear model described tests the hypothesis.
- Lines 182-186: What were these models designed to test?
- Overall in this whole section, I would have liked an explanation of what hypothesis is being tested with each analysis or model. For instance, "if we see X effect, that suggests Y." It was hard for me to follow exactly why each model was important and the subtle distinctions between each set of hypotheses. Adding these explanations would make the links between the analyses and the introduction/discussion clearer and might allay some of the concerns I expressed above.

Results

- Overall, the results were clear. I found the organization into subsections that corresponded with the methods made interpretation easier.

Discussion

- In the introduction, these male sandpipers are described as visiting multiple sites during a given breeding season. This is a key part of defining their movements as nomadic. It becomes clear that these analyses focus on flights from a single location near Utqiagvik; why were subsequent flights not considered?
- The social dynamics introduced in the discussion will be important. Would flying against the wind be beneficial if it means you have fewer competitors? I would have liked some more detail here.
- Line 342: “Individuals changed their heading to compensate for wind drift”: does this counter the idea that birds are just going where the wind is taking them? What does this have to do with your nomadism hypothesis?

Pedler, R.D., Ribot, R.F.H. & Bennett, A.T.D. (2014) Extreme nomadism in desert waterbirds: flights of the banded stilt. *Biology Letters*, 10, 1–5.

Benhamou, S. (2004). How to reliably estimate the tortuosity of an animal's path:: straightness, sinuosity, or fractal dimension? *Journal of Theoretical Biology*, 229, 209-220.

Appendix B

Response to referee comments

Associate Editor, Comments to Author:

Two referees reviewed your manuscript and both express concerns about the lack of statistical support for the claim that flights are opportunistic and driven by wind conditions (ln. 313-315 and 388-389). They also highlight several other important issues that you need to address; notably the role of distribution of females and nests in driving male flight preferences. You need to carefully address all reviewers' concerns in order to demonstrate that your manuscript is novel enough to be published in Proceedings B.

We thank both referees for their constructive and insightful comments. Please, find our response below. The original comment is in black, our response is in blue.

Reviewers' Comments to Author:

Referee: 1

Review of: Wind conditions influence breeding season movements in a nomadic polygynous shorebird

Proceedings of the Royal Society B Johannes Krietsch, Mihai Valcu and Bart Kempenaers

In this paper, Krietsch *et al.* explore the relationships between wind patterns and breeding-season movement patterns in pectoral sandpipers. Using tracking data of male sandpipers, they analyse wind speed (and related metrics) along flight paths with the goal of understanding whether males select breeding sites based on wind patterns (vs. having a more directed goal). Overall, I found the analyses well described and the results clearly presented. However, I found the framing (introduction & discussion) of the manuscript difficult to understand at times, and think the paper should either be re-framed or needs alternative analyses to truly test the hypotheses the authors propose.

We revised the introduction and parts of the methods and discussion to make it easier to understand. Furthermore, we performed the additional proposed analysis creating a null model to test whether males went more often in a direction with favourable winds than expected by chance. See details below.

I agree that the movements of these breeding-season males can be characterized as nomadic because they are irregular. However, I disagree with the authors' statements that movements must be "without a specific geographical target" (line 69) in order to be nomadic. In many cases nomadic movements do appear to be directed (e.g. Pedler *et al.* 2014). In the case given here, nomadic movements could have a known destination *and* use wind support if individuals choose an optimal day to depart. The authors do perform an analysis to test this hypothesis (and reject it), but overall I disagree with the idea that this manuscript tests *whether* sandpipers are nomadic. It was not clear to me whether this issue was primarily conceptual (i.e., a difference in the definition of nomadism) or semantic (i.e., unclear wording).

Thanks for pointing this out. We agree and deleted the sentence about the lack of a target. We also agree that this manuscript is not about whether these movements are nomadic (which is suggested in Kempenaers & Valcu 2017). Rather, in this study, we aim to explore whether and how the wind conditions influence the movements. We revised the introduction to make this clear.

The authors also aim to test for a mechanism of nomadic movements, in other words whether flights are opportunistic in their direction depending on wind. They rely on conclusions such as “most males flew with wind support (line 312)” to support the idea that flights are opportunistic in their direction. In my opinion, a null model would be a more powerful way to test this hypothesis, since it would really answer the question “do more males fly with wind support than would be expected if they had a known destination”? As it is, I do not see a clear quantitative conclusion; for instance, Fig. 3 shows that a substantial proportion of the population flew with average negative wind support. Given the distribution of destination sites and wind speeds, the distribution of wind support *could* be random. For instance, an appropriate null model could randomize departure dates and destinations for each path, then calculate the average wind support and compare that to what birds actually experienced. (Other null models could also be valid, but I wanted to give an example for clarity.)

We thank the reviewer for this excellent suggestion. We added a null model by simulating 10000 random departure dates for each track within the period of actual departures in each season. We then compared the mean wind support of each simulated track with the mean wind support of the real track. The results show clearly that the observed wind support was unlikely a result of males leaving by chance. We added the information to the methods (lines 205-209), results (lines 290-296) and discussion (lines 378-379).

In addition, neither the introduction nor discussion gives enough background on the distribution of females and nests. For instance, where are females present? Are males changing their probability of encountering females by choosing some locations vs. others? This seems like a very important aspect of understanding the nomadic movements, since the benefit of reaching a destination is finding a mate. In the introduction, the authors state that “the costs of reaching a specific site can become part of the quality characteristics of this site.” (Line 71). This is an interesting and straightforward way of thinking about travel costs in movement analyses! However, without other information (i.e., site quality upon arrival), it is hard to draw any conclusion about whether traveling with the wind is actually beneficial. To answer these questions, it would be very helpful to include information on the locations of females, nest success, or other metrics of site quality upon arrival.

We agree that it would be interesting to include information on the locations of females, and on nest success, or other metrics of site quality upon arrival. Unfortunately, we do not have this information. The tracked males went all over the Arctic and there is no way to assess the individual’s reproductive success. The information we do have is that the areas where males became resident are (a) within the known breeding range of the species, (b) snow-free (at least partially) when males arrive (Kempenaers & Valcu 2017). We also know from our local study in Utqiagvik that local mating opportunities are unpredictable and depend on (a) the number of females arriving at the site (note that only 2% of males and <1% of females return to the same site between years), (b) the snow melt pattern (influencing the amount of available habitat over time), and (c) the number and strength of competing males. Consequently, we believe that it is unlikely that males can predict the quality of a site at the scale of their movements. Therefore, we hypothesize that the cost of reaching a particular site (related to wind assistance) should be an important factor in deciding where to go next. We

modified the introduction to clarify this (lines 62-66). It is also possible that the role of wind is indirect, in the sense that female movements could be wind-dependent, and that males simply follow the females. We now make this point in the Discussion (lines 427-429).

As an alternative to these analyses, I think this paper already provides a clear description of the natural history of this system and the flights of pectoral sandpipers. The fact that flights tend to have wind support and that inter-annual differences in flight destinations depend on wind appears to be a new result. In this case, I would think the paper would be more suited to a more specific (e.g. ornithology) journal. I have included a few more detailed comments below.

Introduction:

- Line 47: I wouldn't necessarily characterize this as "higher plasticity," since it is plastic in space but not time.

We agree and reformulated this sentence (line 46).

- Line 50: "individuals might face trade-offs with optimal timing." What does this mean? Tradeoffs between what and what?

We meant to say that there are trade-offs between waiting for favourable wind conditions to benefit from wind support and arriving earlier but spending more energy on flight. We revised the sentence (line 49).

- Lines 43-57: I struggled somewhat to grasp the goal of this paragraph. It seems to aim to bridge the gap between the first paragraph about optimal migration and the third, about nomadic movements, but I'm not sure exactly where the link is. It seems unnecessary as written, so I would suggest removing it or editing it to make the goals more clear.

The point we wanted to make is that species-specific differences in life-history and ecology can constrain how birds can optimally use the wind (described in the first paragraph). We now combined the first and second paragraph and shortened it (lines 42-54).

- Paragraph lines 79-86: This comes a little out of the blue. Can you introduce these concepts of performance as they relate to wind and what the alternatives might be? This section is not my area of expertise, but I am having trouble understanding how it could be increasing wind support *wouldn't* result in lower costs.

We agree that the term "performance" was not clear and we removed it. Indeed, increase in wind support will always lead to lower flight costs. We first describe the magnitude of the wind support prior to discussing its consequences.

Methods

- Line 128: I think this "sinuosity" would be more accurately called "straightness" or "efficiency," since you are not measuring changes in direction or step lengths (Benhamou 2004)

We agree and exchanged sinuosity with straightness throughout the manuscript.

- Movement tracks have data points every 15 minutes, whereas wind data is every 6 hours. Are the results sensitive to the time scale of the movement tracks? This could be important, especially if wind speeds are variable enough within each 6-hour window. I would have liked to see a sensitivity analysis of the effect of coarsening the temporal resolution of the movement data.

Thanks for pointing this out. We now show that our results are not sensitive to the time scale of the wind data, because winds are generally not changing so fast. To show this, we included an analysis where we used the wind data from 6 hours before and from 6 hours after the actual flight and calculated the mean wind support for each track. These 6-hour shifted wind data correlate strongly with the actually used wind data ($r > 0.98$). Consequently, variability within the 6 hour intervals is unlikely to influence our results. We added this sensitivity analysis in the Methods section (lines 96-97).

- Section (g): I need more explanation of this hypothesis! Why would this be true only if individuals are nomadic? I am also not sure how the linear model described tests the hypothesis.

We agree that this hypothesis needed more explanation. Kempnaers & Valcu 2017 argue that males move nomadically and have no final breeding destination or ultimate “goal”, but rather sample multiple breeding sites for different amounts of time. We considered a subset of long-distance flights that started from Utqiagvik. Because these flights were initially over sea, the direction of flight decided where in the breeding range males can start sampling potential breeding sites (the next tenure site is reached soon after reaching land; median 73 km). Even if males would have a far-distance goal somewhere, the wind support would predict where they enter suitable breeding habitat, and it is likely that these males assess the local quality of the site and compete for access to females at any place they stop (tenure site, see Kempnaers & Valcu 2017).

- Lines 182-186: What were these models designed to test?

We added a sentence describing the aim of these models (lines 192-194).

- Overall in this whole section, I would have liked an explanation of what hypothesis is being tested with each analysis or model. For instance, “if we see X effect, that suggests Y.” It was hard for me to follow exactly why each model was important and the subtle distinctions between each set of hypotheses. Adding these explanations would make the links between the analyses and the introduction/discussion clearer and might allay some of the concerns I expressed above.

We revised this section and explained more clearly what hypothesis is tested (see lines 192-220).

Results

- Overall, the results were clear. I found the organization into subsections that corresponded with the methods made interpretation easier.

Discussion

- In the introduction, these male sandpipers are described as visiting multiple sites during a given breeding season. This is a key part of defining their movements as nomadic. It becomes clear that these analyses focus on flights from a single location near Utqiagvik; why were subsequent flights not considered?

We focus on the first long flight from Utqiagvik over sea and did not include subsequent flights, to have comparable data. That is, we consider individuals that leave from the same area within the breeding range and approximately in the same period. Once males reach land, they often sample sites on a smaller geographical scale, perhaps because the breeding season is so short. We revised the description of the aim of the study in the introduction to make this clear (lines 73-76).

- The social dynamics introduced in the discussion will be important. Would flying against the wind be beneficial if it means you have fewer competitors? I would have liked some more detail here.

This is an interesting question, but hard to answer. Not all males fly via Utqiagvik, which means that even if an individual flew against the wind to the Canadian Arctic, it may encounter males that directly migrated north to reach the Canadian Arctic. Moreover, if flying against the wind to avoid competitors would be a strategy, we would also expect to find males flying against the wind to the Russian Arctic. Hence, we argue that the observed pattern might be the result of different costs between flying to Russia vs. the North American part of the breeding area (lines 397-409).

Line 342: “Individuals changed their heading to compensate for wind drift”: does this counter the idea that birds are just going where the wind is taking them? What does this have to do with your nomadism hypothesis?

Indeed, this suggests that most of the observed individuals did not simply fly with the flow (perhaps with the exception of birds that made loop flights). The wind situation over the Arctic Ocean does not allow to only “go-with-the-flow” (see Supplementary Movies), because it would result in loops or in birds flying towards the North Pole. The initial flight direction may be chosen opportunistically, resulting in individuals flying in favourable wind conditions. We added a sentence in the discussion to make this clear (lines 391-393).

Pedler, R.D., Ribot, R.F.H. & Bennett, A.T.D. (2014) Extreme nomadism in desert waterbirds: flights of the banded stilt. *Biology Letters*, 10, 1–5.

Benhamou, S. (2004). How to reliably estimate the tortuosity of an animal's path:: straightness, sinuosity, or fractal dimension? *Journal of Theoretical Biology*, 229, 209-220.

===

Referee: 2

General Comments:

The authors investigated how experienced wind conditions influenced the breeding season movements in a “nomadic” polygynous shorebird. To answer this question, the authors used a subset of previous published migratory tracks of male pectoral sandpipers (hereafter: PeSa) that were

tagged in Barrow, subsequently the authors linked the breeding season movements with the wind-conditions en route and at departure. While I really like the interesting observation and patterns between the different “groups” of migrants (East, West & Loop; Fig. 3), I’m sorry to say that I dislike the current framing of the MS. The main reason for this is the lack of causality and the fact that the authors over-emphasize the effect of wind on breeding season movements (see for instance the large contradiction between lines 388-389 and lines 358-359). Based on the provided data and analysis I’m currently not convinced that the authors provide a solid explanation of the “nomadic” breeding season movements of PeSas (totally wind-driven). Therefore I suggest that authors either perform some additional analysis (see comments below) or change the framing of the MS. For instance, the authors could frame the paper more descriptive/observational and discuss that some movements are only made with enough wind-assistance while others (to the east) are also performed without a strong wind-assistance. The authors could then speculate whether this is caused by potential individual differences; sired offspring at a leck, age, group-composition and wind-conditions.

We thank the reviewer for the positive appraisal of our study and for pointing out the problem with the framing. We revised the introduction to make the aim of this study more clear. We ask in how far wind conditions play a role in where individuals go to sample potential breeding areas, but this does not exclude other factors that might play a role. Our analyses – including a new test against a “null model” (see comment reviewer 1) – show that the local wind conditions influence where males go on a large scale (i.e. Russian vs. North American Arctic) and as a result influence where males sample breeding sites, but it is clear that most individuals do not simply “go with the flow”. See also our response to the detailed comments below.

Also, as the authors aim to understand the general “breeding season movements of PeSas” they should be very careful when excluding certain tracks. With the current sub-setting of tracks, especially rule (iii) in line 105 and the removal of overland tracks in lines 121-122 the authors remove part of the natural existing variation in “breeding season movements of PeSas”. Overland tracks are also part of the breeding season movements and by excluding these tracks you can no longer talk about breeding season movements in general, but only about breeding season movements over sea-ice.

Overland tracks were rare in our data set (4%, 4 of 89 flights). We added this information in the Methods to make clear that we analysed the majority of long flights. For further details about our selection criteria, see below.

Major Comments:

Lines 56-57: Argument behind the comma is not logically true; there could also be variable goals (i.e. nomadism) together with goal-oriented flights, these aren’t opposites as presented here. Especially when individuals have knowledge about the area to the west and the area to the east they can flexibly adjust their goal to this. That would be goal-oriented nomadism! Thus I would suggest to reformulate this last part of an overall well-written introduction.

We agree and reformulated this sentence (lines 51-54).

Lines 61-68. The authors give a very nice description on the ecology of MALE PeSas. However, as the authors mention in line 64-65 these “horny” PeSas males are in search of mating opportunities. This makes the idea of nomadism i.e. “going with the flow” less likely and “going with the females” much more (see also your earlier publication on this idea of “mate-searching” [27]). Consequently it is of great importance to give at least some basic information on the migration ecology of FEMALE PeSas. For instance, is there anything known about their timing of migration in Barrow? Do they arrive later, earlier or at the same time? And could it be that the females are actually the nomadic individuals, while the males just follow the females?

We thank the reviewer for this comment. We agree and added information about females in the introduction (lines 63-67) and discussion (lines 428-430) of the manuscript (see also reply to reviewer 1 on the same topic). Females were present at the study site in Utqiaġvik when males performed the observed flights and indeed it is possible that males followed departing females.

Lines 68-70: True, however this requires sampling the departure decision of individuals multiple times. Since you only have limited repeated measurements of the same individual, it is difficult for you to make generalizations. Especially because one half of the repeated measured individuals does the opposite of the other half you cannot really come to a conclusion on this. Please reframe the entire manuscript accordingly.

We deleted this sentence in the revised manuscript and reframed the introduction to avoid misunderstandings. Unfortunately, it is difficult to get repeated measures of the same individual, because site fidelity is extremely low (2% of males and <1% of females return locally).

Lines 77-78: In my opinion you can only answer this question with repeated tracks of the same individual or by using a more mechanistic modelling approach like Revell & Somveille 2017 Sci. Rep. 7: 9870. With the mechanistic modelling approach you could include all the possible available “goals” when you combine the observed goals with remote sensing and GIS technologies to identify breeding/lecking areas. Then, by combining the wind-conditions and possible goals you could perform a simulation exercise whereby you have 1-million particles leaving from Barrow at a certain moment. By varying the importance of the different components in your model you can then infer which component is the most important and explains the breeding season movements of male PeSas.

This is an interesting suggestion, but we kindly disagree that this approach is useful here. The main issue is that possible available goals are virtually everywhere in the breeding range, because pectoral sandpipers have no fixed lek locations and most of the breeding range contains potentially suitable habitat. The quality of a site is also unpredictable, because of variation in snow cover and snow melt conditions, variation in number of females and variation in number of competitors. However, also in response to a comment by reviewer 1, we now include a formal test to determine whether males leave with wind support more often than expected by chance.

Lines 87-89: Again, you can describe the relationship between wind and your observation (which you do very neatly), but you cannot assess whether your observation depends on this. You should stick to your first aim and discuss what this might mean, but you shouldn't aim to do something you currently cannot.

We revised this sentence (lines 84-85).

Line 105, point (iii): This limits the generalizations you can make about the relation between your observed tracks and wind conditions to those birds that departed over sea. This limitation should be acknowledged in your discussion as it might change the relation you find. You could also add a statement or analysis that choosing for this subset of individuals does not influence your results.

We added the information that this criterion only excluded 4 tracks (4% of flights) to make clear that the presented data consider the majority of, but not all individuals (lines 101-104).

Line 120 (point ii): Same as above, by excluding this data you limit how general your results are. In this case you analyze track lengths that are shorter than actually observed and your found relationship with wind conditions is only valid for this part of the track. I don't mind this, your analysis is still informative, but you must acknowledge the consequences of excluding this data. Best would be to see an analysis with and without the data excluded. That would give additional information about whether flights over sea are in fact more or less dependent on wind than flights over land (with additional cues).

In the majority of cases, the distance flown over land was negligible compared to the distance flown over the ocean. Nevertheless, the point is well taken and we now explicitly mention again in the Discussion that we excluded data of flights over land and what the consequences are (lines xxx-xxx).

Lines 193-194: True, but you show that they flew both with a tailwind and a headwind. And your additional analysis (lines 195 – 201) shows that they didn't wait for favorable winds, instead you might interpret your results as if they flew with a headwind on purpose (as they didn't wait for a tailwind).

This is indeed what we think is happening. We rewrote part of the discussion to make this point clear (lines 394-357).

Lines 228-230: See above, this is only true for those parts of the flight that were over the sea. It might be good to acknowledge that.

We now explicitly state this (lines 247-249).

Lines 245-247: "Optimal" is not the same as the altitude with maximum wind support. I would suggest referring to it as the altitude with maximum wind support as there is no information whether this is also optimal or not. (see also lines: 325-327)

We agree that this can be misunderstood. To avoid confusion we changed the name to altitude with maximal wind support, short "maxWs" and made this change throughout the manuscript.

Lines 252-254: Please supply a statistic that evaluates the Goodness-of-fit of this model.

We present this statistic in Table S2 for the best fitting model (750 m, ground speed) and acknowledge that this was not clear in this form. We added a sentence in the caption of Table S1 to make this link clear for the reader.

Lines 267-272: Is this a significant proportional difference between wind conditions for the different groups?

Yes, the difference in wind support is significantly different between east and west, based on a linear model with mean wind support by track as response variable and category (east, west and loop) and year as fixed factors (using a post-hoc Tukey test to compare within category):

term	estimate	s.e.	statistic	p
category (west - loop)	3.31	1.35	2.46	0.04
category (west - east)	4.33	0.77	5.61	<0.001
category (loop - east)	1.03	1.31	0.78	0.71
year (2014)	0.18	0.74	0.24	0.99

This difference in mean wind support between categories and year can also be seen in Figure 3. We added a sentence to the results and included this table in the supplementary material (Table S8).

Line 305-306: They might also have expended more energy by flying for a longer time, therefore please quantify this, acknowledge this or remove this slightly misleading statement.

We removed the sentence.

Line 309: “seemingly” nomadic is not necessarily the same as a large variation in routes.

We rephrased this sentence (lines 336-337).

Line 352: A constant flight direction makes a goal-oriented flight more likely especially as your results show not all of them “go with the flow” .

We agree that this makes at least a broad goal likely (birds have to reach land). We suggest that the initial flight direction is often chosen opportunistically, resulting in individuals flying in favourable wind conditions to a large-scale goal (e.g. the Russian part of the breeding range). We added a sentence in the discussion to make this point more clear (lines 391-393).

Minor Comments:

Line 73: “how individual pectoral sandpipers respond to wind conditions” ◇ Where? At the breeding grounds? En route? At departure? At arrival?

We deleted this unclear sentence in the revised version of the introduction.

Line 79: “centre of the range” ◇ “centre of their distribution range”

We changed it to “breeding range” (line 78).

Lines 97-99: The authors should mention the continuous duty-cycle of the tags and the number of received Argos-locations per hour.

We added the relevant information (lines 96-97).

Line 112: I would suggest to explicitly mention that these are repeated WITHIN-season tracks and not between-seasons.

We added this, as suggested (line 111).

Lines 131: ~80km: I would provide this information in degrees, the ERA-Interim grid is also in degrees.

Changed as suggested (line 131).

Lines 138-140: According to your results, PeSas fly at an altitude of 750m, thus if they use local wind-conditions for their departure-decision you should at least show that there is no difference between ground-level wind-conditions at departure and wind-conditions at 750m.

In a pair-wise comparison of mean wind support during the first 50 km at 10 m and 750 m the wind support is significantly higher at 750 m (mean difference = 2.66 m/s, $n = 85$, $p < 0.001$), which is due to an increase in wind speed with altitude. However, there is also a strong correlation between the mean wind support during the first 50 km at 10 m and 750 m (Pearson's $r = 0.92$, $n = 85$, $p < 0.001$). Consequently, the wind conditions on the ground are an excellent predictor for the wind conditions at 750 m. We added this analysis to Methods (lines 176-179), Results (lines 273-276) and Discussion (lines 367-379).

Line 164: "a change" does this mean 1 change in altitude per hour? Also, do you allow changes between 0-3000m or just changes between consecutive altitudes?

Yes, it means one change in altitude per hour and the change can be to any of the six altitudes. We changed the sentence to make this clear (lines 164-167).

Lines 329-331: This could be, but the "optimal" temperature zone will be species specific. I would suggest remove this part of speculation and argue that the authors need to measure (e.g. GPS-transmitters) the actual flight-altitude before discussing the flight-altitude.

We modified the sentence as suggested (line 356).

Lines 387-396: After reading the MS I was wondering whether the authors are able to infer the status (alive/dead) of a tagged PeSa (e.g. Sergio et al. 2018) and if so, whether the authors did observe a proportional difference in mortality-rate/tag-loss between the different movements (west, east and loop or amount of wind-assistance). Also, and just out of curiosity, I was wondering whether the authors did clip a small part of the feathers (see also Yohannes et al. 2012 Ibis). It would be very

interesting to see whether the breeding season movements differ between the two different age-classes and whether the interaction with wind differs per age-class.

We thank the reviewer for these interesting suggestions. Unfortunately, we cannot distinguish between death or tag loss because the PTTs were glued on the back of the bird. Thus, a PTT might send its last data while laying on the ground, either because it fell off (most likely) or because it was attached to a dead body.

Indeed, it would be interesting to know more about the past experience of these males. Yohannes et al. (2012) suggests that first-year males do not migrate to the breeding area (Utqiagvik). Consequently, all males presented in this study are most likely at least two years old.

Appendix C

Review of RSPB-2019-2789

Wind conditions influence breeding season movements in a nomadic polygynous shorebird
Krietsch et al.

I found this version of the manuscript easier to follow than the previous version and the new analyses provide better support for the authors' conclusions. I thank the authors for taking the time to implement new analyses and to so substantially rewrite the text. Overall, I agree that these revisions have improved the manuscript, but I do have a few remaining questions and suggestions.

The new introduction is clear, but I would have liked to have seen these ideas feature more prominently in the discussion as well. In particular, the authors now open by discussing optimal migration theory and alluding to the tradeoffs between energy and time limitation in migration. In the discussion, I would be interested to hear more about how the results inform this theory. (E.g., it seems that these results support energy-limitation for this species because it would have been faster to take the shortest route, but that would have produced less wind support.) Since this theory has rarely been applied to nomadism, what do these results suggest about similarities between migrants and nomads? Similarly, the authors make a statement at L73-74 about the relationship between movement costs, unpredictability, and decisions about where to go. Do the results support this idea? The last paragraph touches on these ideas, but I would be interested in broader conclusions throughout the discussion.

I would also like to hear more about other systems and implications of this work. As far as I could tell, the discussion contained few conclusions beyond this system, which I would expect in a journal like Proceedings B. To what other species/systems might these ideas apply? Why is understanding these ideas important (e.g., making predictions, conservation, etc.)?

Minor comments:

Line 50: "overwritten" should be "overridden"

Line 72: "fast" should be "quickly"

Line 122: Please define "tenure site" and how you identified it. This isn't a term I have heard before.

Line 196: "hereafter"

Line 190: What do you mean by "To test whether males covered longer distances over sea before reaching land"? Longer than what?

Line 221: What do you mean by "worse wind conditions"? Is this the same as less wind support?

Line 258: Typo, "and classified as loop flights" doesn't make sense in this sentence.

Line 269: Can you express this as a percentage of the actual wind support? It does seem like a small number but it would be easier to compare is a proportion rather than an absolute value.

Line 412: What are they "overshooting"?

Appendix D

Response to referee comments

Associate Editor Board Member, Comments to Author:

The two reviewers that evaluated the previous version of your ms are satisfied with your revised manuscript. However, you need to make a better case for the general applicability of your results. At the moment, the discussion is too focused on your model species; you need to address how your results inform optimal migration theory. Referee 2 provides excellent suggestions on how to address this problem.

We added a “Conclusions” section with an additional, short paragraph highlighting the general implications of this study (lines 426-454).

Reviewer(s)' Comments to Author:

Referee: 1

I found this version of the manuscript easier to follow than the previous version and the new analyses provide better support for the authors' conclusions. I thank the authors for taking the time to implement new analyses and to so substantially rewrite the text. Overall, I agree that these revisions have improved the manuscript, but I do have a few remaining questions and suggestions.

The new introduction is clear, but I would have liked to have seen these ideas feature more prominently in the discussion as well. In particular, the authors now open by discussing optimal migration theory and alluding to the tradeoffs between energy and time limitation in migration. In the discussion, I would be interested to hear more about how the results inform this theory. (E.g., it seems that these results support energy-limitation for this species because it would have been faster to take the shortest route, but that would have produced less wind support.) Since this theory has rarely been applied to nomadism, what do these results suggest about similarities between migrants and nomads? Similarly, the authors make a statement at L73-74 about the relationship between movement costs, unpredictability, and decisions about where to go. Do the results support this idea? The last paragraph touches on these ideas, but I would be interested in broader conclusions throughout the discussion.

I would also like to hear more about other systems and implications of this work. As far as I could tell, the discussion contained few conclusions beyond this system, which I would expect in a journal like Proceedings B. To what other species/systems might these ideas apply? Why is understanding these ideas important (e.g., making predictions, conservation, etc.)?

Minor comments:

Line 50: “overwritten” should be “overridden”

Corrected (line 48).

Line 72: “fast” should be “quickly”

Changed as suggested (line 70).

Line 122: Please define “tenure site” and how you identified it. This isn’t a term I have heard before.

We changed the term “tenure site” to “residency area”, which is easier to understand and added that this is identified as spatial clusters of points, as detailed in Kempnaers and Valcu 2017) (line 120 and 123).

Line 169: “hereafter”

Corrected (line 169).

Line 190: What do you mean by “To test whether males covered longer distances over sea before reaching land”? Longer than what?

We tested whether males covered longer distances with higher wind support. We added this in the sentence to make it clear (line 189).

Line 221: What do you mean by “worse wind conditions”? Is this the same as less wind support?

Yes. We make this clear now (line 220).

Line 258: Typo, “and classified as loop flights” doesn’t make sense in this sentence.

Deleted.

Line 269: Can you express this as a percentage of the actual wind support? It does seem like a small number but it would be easier to compare is a proportion rather than an absolute value.

We added the wind support of the fixed altitudes as comparison (line 269).

Line 412: What are they “overshooting”?

We added that they overshoot land (suitable breeding habitat) to make it clear (line 415).

Referee: 2

Comments to the Author(s).

Dear Johannes and co-authors,

I’m happy to see this nicely revised manuscript come back and I would like to thank the authors for addressing all comments. In particular I want to point out that that the addition of a null-model as explained in lines 205-209 makes a great attribution and strengthens the story! Furthermore, I appreciate the investment you made by revising the text while taking into account the proposed

suggestions, as a result I believe that the interpretation of the results has become more fair and that all readers are informed on the beautiful complexity of ecology. Altogether, I would cite this manuscript as evidence that wind conditions influence the breeding season movements of male Pectoral Sandpipers.

My compliment on this cool paper on a cool system which I am sure you will be able to finalize in short time!

All the best!
Jelle Loonstra

Minor comments/suggestions

Line 39: Even though I hate it be self-promoting;). You could stress the importance of in-flight wind-conditions by citing our recent paper: Loonstra et al. 2019; Ecol. Lett.

Thanks for pointing this out. We added a reference to this recent paper which shows that the wind conditions influence mortality in Black-tailed Godwits.

Line 43: Personally I really like the Honey Buzzard story of Vansteelant et al. 2016 J. Anim. Ecol. as a good example of a bird adjusting its route with respect to the most favorable winds.

We added the suggested reference.

Line 75: missing "." after [30].

Corrected (line 74).

Line 344: Suggestion: "they could potentially reproduce" \diamond "they arrive and potentially reproduce"

Changed as suggested (line 346).

Line 346: To prevent any confusion, it might be good to add: "within the same season" after "long flights".

Added as suggested (line 348).

Line 429: "which themselves sample multiple breeding sites": To make it even more complicated;) But very interesting and cool interesting to have these female tracks!

Lines 430-433: You might want to put this in the perspective of the findings from Kranstauber et al. 2012 Ecol. Lett.

Included as suggested (line 438).

Line 449: Raymond \diamond R and Line 479: RG \diamond RHG

Thanks for pointing this out. Corrected.